# Uptake and Completion of Tuberculosis Preventive Treatment Using 12-Dose, Weekly Isoniazid–Rifapentine Regimen in Bangladesh: A Community-Based Implementation Study

**DOI:** 10.3390/tropicalmed9010004

**Published:** 2023-12-20

**Authors:** Md. Toufiq Rahman, Farzana Hossain, Rupali Sisir Banu, Md. Shamiul Islam, Shamsher Alam, Abu Jamil Faisel, Hamid Salim, Oscar Cordon, Pedro Suarez, Hamidah Hussain, Tapash Roy

**Affiliations:** 1Interactive Research and Development, Bangladesh (IRD Bangladesh), Dhaka 1212, Bangladesh; farzana.hossain@ird.global (F.H.); shamsher.alam@ird.global (S.A.); abujamil.faisel@ird.global (A.J.F.); tapash.roy@ird.global (T.R.); 2Innovations & Grants Team, Stop TB Partnership, 1218 Geneva, Switzerland; 3National Tuberculosis Control Program, Directorate General of Health Services, Ministry of Health and Family Welfare, Dhaka 1212, Bangladesh; dr.rupalisisir298@gmail.com (R.S.B.); dfsalimhamid@gmail.com (H.S.); 4Directorate General of Health Services, Ministry of Health and Family Welfare, Dhaka 1212, Bangladesh; sadyshamiul1@gmail.com; 5Challenge TB Project, Management Sciences for Health, Dhaka 1212, Bangladesh; ocordon@msh.org; 6Action Against Hunger, New York, NY 10004, USA; 7Management Sciences for Health, Arlington, TX 22203, USA; psuarez@msh.org; 8IRD Global, Singapore 048581, Singapore; hamidah.hussain@ird.global

**Keywords:** TB preventive treatment, UNHLM, End TB, TPT, 3HP, Bangladesh

## Abstract

Background: The United Nations high-level meeting (UNHLM) pledged to enroll 30 million in tuberculosis preventive treatment (TPT) by 2022, necessitating TPT expansion to all at tuberculosis (TB) risk. We assessed the uptake and completion of a 12-dose, weekly isoniazid–rifapentine (3HP) TPT regimen. Methods: Between February 2018 and March 2019 in Dhaka, community-based TPT using 3HP targeted household contacts of 883 confirmed drug-sensitive pulmonary TB patients. Adhering to World Health Organization guidelines, contacts underwent active TB screening before TPT initiation. Results: Of 3193 contacts who were advised health facility visits for screening, 67% (*n* = 2149) complied. Among these, 1804 (84%) received chest X-rays. Active TB was diagnosed in 39 (2%) contacts; they commenced TB treatment. Over 97% of 1216 contacts began TPT, with completion rates higher among females, those with more education and income, non-slum residents, and those without 3HP-related adverse events. Adverse events, mainly mild, occurred in 5% of participants. Conclusions: The 3HP regimen, with its short duration, self-administered option, and minimal side effects, achieved satisfactory completion rates. A community-focused TPT approach is feasible, scalable nationally, and aligns with UNHLM targets.

## 1. Introduction

Tuberculosis (TB) infection (TBI) is an important reservoir of TB disease, and it is important to treat individuals with TBI to break the chain of transmission and prevent the further spread of the disease in the community [1]. It is estimated that over one-fourth of the world’s population is infected with *Mycobacterium tuberculosis*, the bacteria responsible for TB disease [2]. If TB infection remains untreated, 5–10% of individuals with TBI develop active TB in their lifetime, with 50% developing the active disease within two years after infection [3,4]. A study by Dye et al. showed that to meet World Health Organization (WHO) End TB targets, TBI treatment will need to be incorporated into TB programs, as merely treating the active disease will not result in a significant reduction in the burden given the large number of individuals infected with TB. These individuals with TBI will continue to give rise to TB cases because of reactivation [5]. The WHO recommends that TB preventive treatment (TPT) should be prioritized for contacts of TB patients [6]. Thus, the prevention of new TB infections and their progression to active TB disease is critical to reducing the burden of the disease and resulting deaths, as well as to achieving the United Nations high-level meeting (UNHLM) on TB and End TB Strategy targets for 2022 and 2030/2035.

TB remains a major public health concern in Bangladesh, with an estimated 360,000 people who developed TB and 44,000 died from TB in 2020 in the country [7]. The country’s annual report on TB shows that, although TB case detection and notification in Bangladesh is increasing steadily (approximately 5% annually), the estimated TB incidence has remained static—between 225/100,000 and 221/100,000 since 2001 [8]. The current National Tuberculosis Control Program (NTP) in Bangladesh focuses heavily on detecting new TB cases and treating these new patients with a limited focus on TPT. Children younger than five years and people living with human immunodeficiency virus (HIV) are prioritized for isoniazid preventive treatment (IPT) with a daily dose for six months. Although the NTP and partners have been implementing IPT among children of <5 years for years, the coverage is only 51% among all eligible children, with a completion rate of around 75% [7]. In Bangladesh, the implementation of TPT strategies is hindered by limited resources that affect healthcare capacity and cultural barriers such as health literacy that affect the public perception of TPT and adherence to it. Global evidence also suggests that the acceptance and completion rates of IPT are often low (30–64%) because of the long duration of treatment [9], and implementation of TPT is challenging. The current initiatives are not enough to achieve a significant reduction in TB infection in line with the UNHLM and End TB targets for TPT. 

The recent WHO guideline recommends several shorter regimens, which can minimize the burden on both patients and health systems. One of the recommended TPT regimens is a once-weekly dose of rifapentine and isoniazid for three months (3HP). This regimen has comparable adverse events and better treatment completion rates [10,11,12,13,14,15,16,17]. The experience with this regimen in low-resource programmatic settings is lacking, but given once-weekly dosing and higher completion rates, as observed in other trials, it is expected to improve adherence and address the operational challenges associated with IPT. At present, there are no data available in Bangladesh on the eligibility, initiation, and completion of TPT for child contacts and the feasibility of TPT for other household (HH) contacts. Understanding this implementation feasibility can potentially inform the effective development of future TPT programs and thus warrants comprehensive programmatic research. The current study was conducted to assess the uptake and completion of the 3HP regimen and better understand the programmatic challenges with the intervention delivery, uptake, and completion of the 3HP regimen for TPT.

## 2. Materials and Methods

### 2.1. Ethical Approval

Ethical approval for the study was obtained from the Bangladesh Medical Research Council (Registration Number 127 14 06 2018). Written informed consent for participation was obtained by project staff prior to enrollment into the study from adults and from guardians/parents of children. All procedures performed in this study involving human participants were in accordance with the ethical standards of the institutional and/or national research committee. All participants enrolled in this study received free services, including costs for investigation and TPT with 3HP.

### 2.2. Study Design and Location

A community-based implementation study was conducted under routine programmatic settings in Dhaka (South City Corporation Zones 3, 4, and 5), Bangladesh, between February 2018 and March 2019 to assess the uptake and completion of TPT with 3HP. The design of this study and the intervention packages were supported by a prior qualitative study on the barriers to and facilitators of existing IPT program in Bangladesh [18]. 

### 2.3. Study Population

The study population comprised the HH contacts of bacteriologically confirmed (B+) pulmonary-drug-susceptible TB (DS-TB) patients (index patients) enrolled for TB treatment at all 12 NTP-linked treatment facilities in the selected areas during the study period. 

### 2.4. Inclusion and Exclusion Criteria

All HH contacts of index DS-TB patients were considered eligible if (i) both the index DS-TB patients and their families lived in our selected study sites and (ii) the individual was aged over two years. The HH contacts were excluded from the study for the following reasons: (i) they were already receiving TB disease treatment or IPT at the time of the HH contact investigation; (ii) they were women who were pregnant or planning pregnancy during the study period; (iii) the HH contact was under two years of age, as rifapentine is not recommended for this age group. Pregnant women were excluded given the limited safety data on the 3HP regimen during pregnancy. This precaution, reflecting ethical considerations and the need to mitigate unknown fetal health risks, aligns with the principle of safeguarding participant safety in clinical research, especially in populations with insufficiently studied treatment effects.

### 2.5. Identification of HH Contacts for TPT

Field Supervisors (FSs) who had prior experience working with TB were recruited by the study and received training on the study implementation process. The FSs extracted the contact details of the index TB patients from the treatment registers and interviewed each index patient over the phone to enumerate the HH contacts. They encouraged the index patient to bring their HH contacts to the nearest health facility for contact screening and clinical evaluation. Once HH contacts arrived at the facility, Treatment Counselors (TCs) counseled the index TB patients and the HH contacts on the risk of developing TB and the importance of TPT. Then, the TCs conducted verbal screening for the presence of TB symptoms, and a physician conducted a clinical assessment. Based on the advice of the physician, free chest X-rays (CXRs) were offered to all contacts to rule out TB disease as per the NTP protocol. Rapid molecular tests (if sputum was available) were performed for the contacts with abnormal CXRs to detect active TB disease. When active TB disease was confirmed, the project field staff connected the patient with an NTP-linked TB treatment facility for treatment initiation. Those HH contacts considered not to have active TB disease were counseled by the treatment counselor and invited to initiate TPT with a 3HP regimen. For HH contacts with self-reported previous episode(s) of TB, the NTP records were reviewed to verify previous treatment and outcome history. If the individual had not completed TB treatment, was not declared cured/had not completed treatment, or did not have documented proof, he or she was referred to the TB treatment facility for further evaluation before being fully eligible for TPT.

In instances where the HH contacts did not come for evaluation or were reluctant to come within one week of the initial phone call, the TCs made at least three reminder phone calls, and then, community health workers (CHWs) from the NTP visited the HHs within three weeks to talk with the contacts face-to-face and motivated them to go for evaluation.

### 2.6. Initiation of TPT with 3HP

The HH contacts who were willing to participate and met the inclusion criteria were considered eligible. The eligible HH contacts took the recommended first dose of the TPT in front of the physician within seven days of the initial evaluation. The 3HP regimen was used among eligible contacts >2 years old and as a self-administered treatment procedure by the participants with support from the project team and the CHWs [11,12].

### 2.7. Treatment Support and Monitoring of Adverse Events

A trained CHW (acting as a treatment supporter) from a local non-governmental organization (NGO) visited the participant at home bi-weekly to follow up on treatment progress and to assess any adverse drug reactions. The treatment adherence was assessed through self-reported pill intake by the HH contacts over the phone and reconfirmed by pill count and reviewing the household diaries of CHWs during follow-up household visits. The treatment completion was defined as completing at least 11 doses of 3HP within 3 months. The TCs called the HH contacts on TPT every two weeks during the treatment period. In addition, the FSs visited all HHs monthly, quantified adherence, asked about adverse events, and recorded the results. If either CHWs or FSs identified a possible adverse event, they immediately communicated with the physician and referred the participant to the hospital for clinical evaluation if needed. Contacts on 3HP visited the health facility every month for follow-up evaluation by the physician. Enablers (USD 36 per month) to promote treatment adherence were provided to CHWs, and travel and investigation costs were reimbursed to the participating families. The HH contacts could also self-report any adverse events to the CHWs or to the project physician, which were recorded on a standardized open-ended adverse event reporting tool. The reported adverse events were immediately assessed, graded, and managed by the physicians. In cases of drop-out from the TPT, the CHWs explored the reason for discontinuation and recorded the reasons.

### 2.8. Data Analysis

All data were analyzed using the Statistical Package for Social Sciences, version 24 (SPSS Inc., Chicago, IL, USA). Descriptive statistics were used to report the data. The data involving continuous variables (age, schooling, income, etc.) were analyzed using a rank-sum test, and the results are presented as the median value plus the minimum and maximum values. Sex, residence, occupation, comorbidity, etc., variables are presented as frequency and percentage. The cumulative probability of an HH contact completing all stages of the TB preventive care cascade was assessed in all eligible participants, and the proportion of HH contacts completing 3HP was assessed among all those who initiated the regimen. Univariate logistic regression analysis was used to analyze the relationships between various factors and the subjects’ completion of TPT, and the results were calculated as an odds ratio (OR) and its 95% confidence interval (95% CI). A multivariable non-conditional logistic regression analysis was then performed on univariate variables that were statistically significant, with the criteria for inclusion being a *p*-value ≤ 0.050. 

## 3. Results

### 3.1. Demographic and Clinical Characteristics 

During the study period, 3193 HH contacts of 883 index TB patients were enumerated and counseled to visit health facilities for evaluation. Of 3193 HH contacts, 67% (*n* = 2149) showed up at the health facilities and were verbally screened, of whom 54% (*n* = 1167) were female, and the mean age of the contacts was 21.2 years (Standard Deviation (SD) ± 17.5). Of the 1216 contacts initiated into TPT, the mean age was 27.4 (SD ± 23.8), and 56% (*n* = 675) were female. Diabetes was the predominant comorbidity among contacts. It was observed that most of the contacts needed two phone calls from the project team to get them to the facilities for evaluation (Table 1).

### 3.2. Evaluation, Initiation, and Completion of TPT among HH Contacts

Of the 2149 contacts who visited the health facilities and were verbally screened, 1804 contacts (84%) completed an evaluation with CXRs. Among CXRs performed, 39 (2%) contacts were found to have active TB (abnormal CXR and molecular test), and 92 (5%) were below two years and initiated on IPT. The remaining 1673 (93%) eligible contacts were invited to enroll on TPT with the 3HP regimen. Of those eligible, 1216 (73%) initiated the 3HP regimen, 97% (*n* = 1175) of whom completed the regimen (Figure 1). The TPT completion among adults was 99.8%, and in females, 99.9%. The cumulative probability of completing all steps of the TB prevention cascade was 63% among the study population.

Among the HH contacts who did not complete the treatment, 78% (*n* = 32) dropped out between 2 and 5 weeks after the treatment initiation. The most common reason for noncompletion of TPT was a refusal to continue treatment due to side effects or fear of side effects (*n* = 32; 78%). Furthermore, 14.6% (*n* = 6) migrated to a different area where 3HP intervention was not available. Only a small proportion (*n* = 3; 7.3%) was advised by a doctor to stop treatment because of unusual events experienced during TPT (Table 2).

### 3.3. Reported Adverse Events 

During the TPT, 5.3% (*n* = 65) of HH contacts on TPT experienced adverse events. However, no major adverse events were observed, nor were any hospitalizations required. Most of the adverse events were Grade 1, and symptomatic management alone resolved the presenting issues (Table 2). The majority of adverse events were reported in subjects aged 15 years and older (*n* = 51; 4.2%) and among females (*n* = 35; 2.9%). Of the three contacts who stopped TPT as per the suggestion of the physician, one was hypertensive and had poor adherence to antihypertensive medications, and two female patients had gynecologic symptoms (spotting). 

### 3.4. Factors Associated with TPT Completion with the 3HP Regimen 

Table 3 presents bivariate and multivariable logistic regression models examining the association between TPT completion with 3HP and demographic characteristics and clinical factors. The multivariable model reveals that TPT completion was higher in contacts aged 15 years or more (OR 1.5; 95% CI 1.1–2.0; *p* 0.043); female contacts (OR 1.7; 95% CI 1.3–2.1; *p* 0.009); contacts with higher education (OR 1.4; 95% CI 1.1–1.9; *p* 0.044); contacts with high HH income (OR 1.5; 95% CI 1.0–2.1; *p* 0.047); contacts those with no comorbidities (OR 1.7; 95% CI 1.1–2.2; *p* 0.046); and those who did not experience any adverse events while on the 3HP regimen (OR 1.6; 95% CI 1.2–2.1; *p* 0.009). 

The multivariable logistic regression model identified the significant independent predictors of TPT completion with the 3HP regimen. Female sex, higher schooling, higher income, older age, contacts with no comorbidities, and contacts who did not experience any adverse events while on a 3HP regimen were all found to be independent predictors of TPT completion with a 3HP regimen. 

### 3.5. Intervention Approaches

We used multiple intervention approaches for contact investigation and TPT enrollment during treatment. They included phone calls only, phone calls plus counseling, and phone calls plus counseling plus home visits. Among those who attended the health facility (*n* = 2149), 88% (*n* = 1890) came based on the phone calls made to the index patients by the project staff; 9.1% (*n* = 196) came based on phone calls plus counseling; and the remaining 2.9% (*n* = 63) came as a result of the combined efforts of phone calls, counseling, and HH visits. Further, during the treatment, reminder phone calls to inform subjects about the next dose schedule, phone-based and/or face-to-face treatment counseling, and follow-up home visits by the project staff and NGO CHWs were all found to be associated with the completion of the 3HP regimen (Table 3).

## 4. Discussion

This is the first population-based study in Bangladesh that has assessed the implementation feasibility of TPT under routine programmatic settings with a 12-dose, weekly 3HP regimen among the HH contacts of DS-TB patients. The use of 3HP for the treatment of TBI was found to be feasible and well accepted. Consistent with the findings of other studies, we found that a weekly 3HP regimen has higher treatment completion rates (97.3%) compared with under-five children in the ongoing IPT program (74.6%) in Bangladesh, with fewer adverse events, and the results were similar across subgroups of people without HIV [19,20,21].

Several other studies that included shorter TPT regimens have shown a better completion rate compared with longer regimens [11,12,22]. The treatment completion rates in this study were also higher than in other large randomized controlled studies, phase 4 studies, and other cohort studies conducted among adults and children in developed and developing country settings [11,12,22,23,24,25,26,27,28,29,30,31,32]. A recent prospective cohort study conducted on children and adolescents in Pakistan using a 1HP regimen reported a 94% completion rate [33]. 

In Bangladesh, the national TB program provides IPT to children aged under five years who are contacts of B+ TB patients, and there is no provision of TPT for adult contacts. The majority (about 62%) of the HH contacts who were enrolled for the TPT in our study were adults, and treatment completion rates among the adults were also as high as children. Our study generated solid evidence that it is possible to implement TPT among adult populations through the existing routine TB program in Bangladesh. We observed that a higher percentage of our study population (56%) were female, and TPT completion was significantly higher among the female participants. This result will help in shaping the healthcare-seeking behavior of female HH contacts. 

We noted a low frequency (5.3%) of adverse events with 3HP among the study participants, mostly of mild severity, and they were comparable with previous studies [22,24,34,35,36]. A recent systematic review including data from 23 randomized and 55 non-randomized studies also reported a similar low frequency of adverse events with 3HP compared with INH monotherapy [36]. The low adverse events observed in our study might also be attributed to the low drop-out and high treatment completion rates among the contacts who initiated the TPT with 3HP.

The project’s field activities were carried out by its project field staff and NGO CHWs alongside their regular community health interventions. The data indicated that intervention approaches resulted in a high level of TPT enrollment, adherence to treatment, and TPT completion. The possible explanations for this high participation and completion rate may be related to the use of multiple approaches like counseling, phone calls only, phone calls plus counseling, and phone calls plus counseling plus home visits. During the treatment period, we also used reminder phone calls to inform subjects about the next dose schedule, phone-based and face-to-face treatment support and counseling, and follow-up home visits—all of these may have had cumulative effects on the outcomes and helped the participants to make informed decisions to adhere to and complete the TPT. Moreover, the CHWs used in this study were involved in community mobilization, and they were well accepted and trusted by the community, which created an enabling environment for the target community [37]. The study demonstrated that it is feasible to implement a TPT intervention utilizing ongoing TB program infrastructure and the facilities of the government and NGOs in a resource-limited setting. 

The prevention of active TB disease with TPT is a critical component of the WHO’s End TB Strategy. This study proved that a convenient and easy-to-administer TPT regimen should be considered to achieve END TB and UNHLM targets. Considering the high completion rate of this TPT, the abundant care in managing HH contacts (evaluation and enrollment to TPT) and the interventions used to support the TPT can be adopted by the national program. However, the NTP should also consider critical issues with the programmatic scale-up of TPT, including policy considerations, ruling out active TB, diagnostic tests and evaluation, the time to start treatment, safety, uninterrupted drug supplies, treatment adherence monitoring, recording, reporting, etc., before adopting a countrywide TPT imitative. Utilizing CXR as an initial diagnostic tool is also critical given its heightened sensitivity, even though our study did not achieve complete coverage. This experience underscores the importance of advocating for CXR usage in early TB detection, particularly in identifying subclinical or asymptomatic patients. 

One of the limitations of this study was that it only implemented TPT in the urban settings of Dhaka, and it is essential to recognize that the experience may differ in rural areas. This may not represent the entire country and limits the generalizability of our results. Specifically, healthcare infrastructure, community dynamics, and patient behaviors may vary in rural settings. However, considering that urban TB is the most challenging aspect of TB control efforts and the presence of extensive community programs (the Sasthya Sebika model) in the rural areas of Bangladesh [38], we believe this study will help in formulating appropriately targeted measures and future TPT programs in Bangladesh. Future research efforts should aim to assess the feasibility and implementation of TPT programs in rural areas considering the specific challenges and opportunities posed by these contexts. The study also did not have a true comparative group for the HH contacts for whom TPT was initiated using 3HP. Future research should aim to explore the effectiveness of the 3HP regimen across various demographic groups and settings. Additionally, comparative studies between the 3HP regimen and the standard 6-month IPT regimen would provide valuable insights into the relative benefits of these approaches. Furthermore, investigations into the long-term outcomes and cost-effectiveness of the 3HP regimen would be beneficial for informing policy decisions and optimizing TB prevention strategies. Further, the study did not use any test to confirm the presence of TBI. The Interferon-Gamma Release Assay is not used in this country, so the tuberculin skin test (TST) was the only test we could use. However, mass, population-wide TBI testing was not feasible considering the high TB burden setting, and the TST is inconclusive. The absence of TBI confirmation in our study could potentially result in some misclassification of TBI status, and this may impact the generalizability of our results to settings with routine TBI testing. In such settings, the decision to initiate TPT might be guided by a combination of clinical evaluation, TBI test results, and individual risk factors. Therefore, the applicability of our findings to areas with robust TBI testing should be made with an awareness of this limitation.

## 5. Conclusions

In this community-based implementation study, TPT using 3HP was found to have a high completion rate. The convenient weekly regimen of 3HP, the shorter treatment duration, and minimal adverse events resulted in higher treatment adherence among those who were enrolled in this study. The study findings exhibited a strong case for considering the integration of the 3HP regimen into the national TB control program in Bangladesh. The study demonstrated that the identification of potential HH contacts for TPT in urban areas and high treatment completion could be achieved through a well-designed, community-based program using the existing program structure and involving appropriately trained CHWs. Targeted awareness creation, counseling, rigorous follow-up, and a self-administered TPT option also contributed to achieving higher TPT adherence.

## Figures and Tables

**Figure 1 tropicalmed-09-00004-f001:**
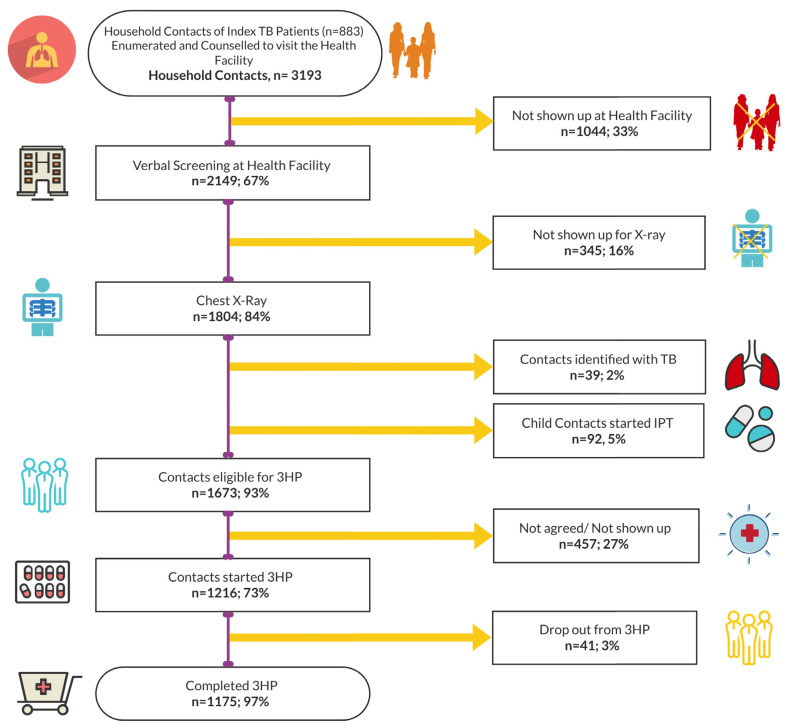
TPT enrollment and completion cascade with 3HP, February 2018–March 2019.

**Table 1 tropicalmed-09-00004-t001:** Socio-demographic characteristics of the household contacts of index TB patients who were verbally screened (*n* = 2149) and initiated TB preventive treatment (*n* = 1216) using the 3HP regimen.

Variables	Household Contacts Verbally Screened (*n* = 2149)	Household Contacts Who Initiated TPT with 3HP (*n* = 1216)
Age—mean age (± SD)	21.2 (±17.5)	27.4 (±23.8)
<5	65 (3.0%)	40 (3.3%)
5–<15	484 (22.5%)	272 (22.4%)
15 and above	1600 (74.5%)	904 (74.3%)
Female	1178 (54.8%)	675 (56.0%)
Schooling in year		
No schooling	323 (15.0%)	146 (12.0%)
1–5 years	726 (33.8%)	383 (31.5%)
6–10 years	710 (33.0%)	345 (28.4%)
10+ years	390 (18.1%)	342 (28.1%)
Household income/month in BDT (mean ± SD) *	14,532 ± 9853	15,251 ± 10,235
≤5000	14 (0.7%)	7 (0.5%)
5001–10,000	321 (14.9%)	165 (13.6%)
10,001–20,000+	1814 (84.4%)	1044 (85.9%)
Original residence		
Permanent resident of Dhaka	677 (31.5%)	398 (32.7%)
Tenant (rural-to-urban migrant)	1472 (68.5%)	818 (67.3%)
Current dwelling status		
Living in slums	1027 (47.8%)	589 (48.4%)
Non-slum households	1122 (52.2%)	627 (51.6%)
Occupation		
Student and dependent child	543 (25.3%)	361 (29.7%)
Day labor/garments/factory work	605 (28.2%)	234 (19.2%)
Self-employed and business	359 (16.7%)	138 (11.3%)
Public/private service	246 (11.4%)	132 (10.9%)
Homemaker	380 (19.9%)	341 (28.0%)
Unemployed	16 (0.7%)	10 (0.8%)
Current or past smoker	892 (41.4%)	483 (39.7%)
Comorbidity		
No comorbidity	1372 (68.5%)	1194 (98.2%)
Diabetes mellitus	95 (4.4%)	16 (1.3%)
Hypertension	409 (19.0%)	04 (0.3%)
Asthma	14 (0.7%)	01 (0.1%)
Thyroid dysfunction	159 (7.4%)	01 (0.1%)
Experienced any adverse events	NA	65 (5.3%)

* USD 1 = BDT 110.

**Table 2 tropicalmed-09-00004-t002:** Number and percentage of household contacts with adverse events after any 3HP dose by type and grade (*n* = 65).

Experienced Adverse Events after Any 3HP Dose by Type *	Frequency ** with Grading ***			
	Not Graded	Grade 1	Grade 2	Grades 3, 4, and 5
Gastrointestinal—nausea/vomiting	-	20 (1.64%)	01 (0.08%)	-
Neurological symptoms	-	01 (0.08%)	-	-
Muscle pain	-	02 (0.16%)	-	-
Hepatotoxicity	-	00 (0.00%)	-	-
Flu-like symptoms	-	10 (0.82%)	01 (0.08%)	-
Dermal—itching/skin rash	01 (0.08%)	09 (0.74%)	-	-
Respiratory symptoms	-	02 (0.16%)	-	-
Fatigue	-	07 (0.58%)	-	-
Headache	-	06 (0.49%)	-	-
Other symptoms ‡	-	05 (0.41%)	-	-
Total experiencing adverse events	01 (0.08%)	62 (5.09%)	02 (0.16%)	00 (0.00%)

* Includes all adverse events reported by the participants deemed to be “related” or “possibly related” to 3HP medication by the project physician. ** Number of participants experiencing an adverse event of a certain severity with one or more doses where each participant is counted only once at the highest level of severity for that type, and the percentages are taken out of the total number of patients on TPT. *** Grade 1 = mild—discomfort noticed but no disruption of normal daily activity; Grade 2 = moderate—discomfort sufficient to reduce or affect daily activity; Grade 3 = severe—inability to work or perform a normal daily activity; Grade 4 = life-threatening or disabling—represents an immediate threat to life; Grade 5 = death—death related to an adverse event. ‡ Other related or possibly related adverse events included dizziness, insomnia, increased blood pressure, and gynecologic symptoms (spotting) (all Grade 1).

**Table 3 tropicalmed-09-00004-t003:** Bivariate and multivariable logistic regression examining the association between TPT completion with 3HP and demographic characteristics and clinical factors.

Variables	Bivariate Model		Multivariable Model *	
	OR (95% CI)	*p* Value	OR (95% CI)	*p* Value
Current age (r = age < 15 years)	2.1 (1.8–2.4)	0.004	1.5 (1.1–2.0)	0.043
Female (r = male)	2.5 (1.7–2.8)	0.002	1.7 (1.3–2.1)	0.009
Schooling in year (r = 0)	2.2 (1.7–2.5)	0.004	1.4 (1.1–1.9)	0.044
Monthly household income > BDT 10,000 (r = ≤ BDT 10,000) ^a^	1.7 (1.3–2.1)	0.005	1.5 (1.0–2.1)	0.047
Permeant urban resident (r = rural-to-urban migrant with temporary settlement)	1.2 (0.8–2.3)	0.098	NA	
Lives in non-slum household (r = lives in slum)	1.6 (1.3–2.1)	0.049	1.1 (0.7–2.4)	0.088
Occupation: non-manual work (r = manual work)	1.1 (0.7–2.7)	0.106	NA	
No comorbidities (r = have had any comorbidities)	2.3 (1.8–2.5)	0.008	1.7 (1.1–2.2)	0.046
Experienced no adverse events (r = experienced any adverse events)	1.8 (1.3–2.4)	0.003	1.6 (1.2–2.1)	0.009
Intervention Approaches				
Reminder phone calls + treatment counseling (r = reminder phone calls only for the next dose schedule)	2.7 (2.1–3.2)	0.002	1.9 (1.5–2.4)	0.007
Reminder phone calls + treatment counseling + follow-up home visit by health workers (r = reminder phone calls only)	3.2 (2.7–3.6)	0.001	2.1 (1.5–2.7)	0.005

* The multivariable model only included the variables that were found to be statistically significant in the bivariate model, with the criteria for inclusion being a *p*-value ≤ 0.050; ^a^ USD 1 = BDT 105; r = reference category; OR, odds ratio; CI, confidence interval.

## Data Availability

All data are fully available without restriction. However, data cannot be shared publicly because the detailed data were generated under programmatic conditions. The data are available from the National TB Control Program (directormbdc@gmail.com) for researchers who meet the criteria for access to confidential data.

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
