# Peer review of "Uptake and Completion of Tuberculosis Preventive Treatment Using 12-Dose, Weekly Isoniazid–Rifapentine Regimen in Bangladesh: A Community-Based Implementation Study"

_tropicalmed, 2023, doi:10.3390/tropicalmed9010004_

Round 1

Reviewer 1 Report

Comments and Suggestions for Authors

To ascertain the acceptance, initiation, uptake and completion of the 3HP regimen was beneficial to understand anti-tubercular treatment. In this study, the authors investigated the uptake and completion of isoniazid-rifapentine (3HP) TPT regimen from February 2018 to March 2019 in Dhaka. The results showed the 3HP weekly regimen with minimal side effects and short duration, achieved satisfactory completion rates in Dhaka. However, to make this manuscript suitable for publication, I suggest its publication after minor revisions.

1. The abbreviations should be in full form when it first appears in the article.

2. Some incorrectly formatted existed in Table 1, Table 2, and Table3. Please carefully checked it.

3. Please insert correct volume and pages, and check the writing format to follow the reference style of the journal.

Author Response

Reviewer #1: To ascertain the acceptance, initiation, uptake and completion of the 3HP regimen was beneficial to understand anti-tubercular treatment. In this study, the authors investigated the uptake and completion of isoniazid-rifapentine (3HP) TPT regimen from February 2018 to March 2019 in Dhaka. The results showed the 3HP weekly regimen with minimal side effects and short duration, achieved satisfactory completion rates in Dhaka. However, to make this manuscript suitable for publication, I suggest its publication after minor revisions.

Thank you for acknowledging the significance of our study in understanding the uptake and completion of the isoniazid-rifapentine (3HP) Tuberculosis Preventive Treatment (TPT) regimen. We appreciate your constructive feedback and are pleased to hear that our manuscript is considered suitable for publication following minor revisions. We have carefully addressed your suggestions to further refine the manuscript for publication. Your feedback is invaluable in helping us improve the clarity and impact of our study.

  1. The abbreviations should be in full form when it first appears in the article.

Thank you for pointing out the need to clarify abbreviations. We have revised the manuscript to ensure that all abbreviations are spelled out in full at their first occurrence. This update has been applied consistently across all sections of the manuscript to enhance clarity (Lines -20, 22, 25, 43-44, 48, 61, 81, 135, and 201).

  1. Some incorrectly formatted existed in Table 1, Table 2, and Table3. Please carefully checked it.

Thank you for pointing out the formatting issues in Tables 1, 2, and 3. We have carefully reviewed and corrected these tables to ensure an accurate and clear presentation of the data. We appreciate your guidance in this matter.

  1. Please insert correct volume and pages, and check the writing format to follow the reference style of the journal.

Thank you for the reminder to adhere to the journal's reference style. We have revised the references to ensure correct volume and page numbers, and their format is now consistent with the journal's guidelines.

Reviewer 2 Report

Comments and Suggestions for Authors

Abstract

Clarity and Conciseness: The abstract efficiently outlines the study's key aspects but includes some repetitive details, such as the specifics of the regimen and the study's setting. For instance, mentioning "12-dose, weekly isoniazid-rifapentine (3HP) TPT regimen" multiple times could be streamlined for brevity.

Introduction

Depth of Background Information: While the introduction highlights the global and national TB context, it could be improved by discussing specific challenges in Bangladesh, such as resource limitations or cultural barriers to TB prevention.

Rationale for the Study: The introduction explains the choice of the 12-dose regimen but lacks comparative data on its efficacy against other regimens used in Bangladesh. A scenario where another regimen may have been less effective in a similar setting could illustrate the need for this study.

Methods

Study Design and Population: The method section is comprehensive but lacks detailed justification for excluding specific groups like pregnant women. Elaborating on the potential risks or ethical considerations for these exclusions would enhance understanding.

Data Collection Process: The manuscript details participant enrollment and treatment monitoring. However, it could describe the specific methods used to ensure adherence, such as digital tracking tools or patient diaries.

Results

Presentation of Findings: The results section clearly presents completion rates and demographic data. Providing confidence intervals for these rates would offer a more precise understanding of the data's reliability.

Subgroup Analysis: The manuscript indicates different completion rates across demographics but doesn't delve into why these differences might exist. Analyzing factors like socioeconomic status or education level could provide deeper insights.

Discussion

Comparative Analysis: While the discussion puts the study in context, it lacks a detailed comparison with other studies using different TB treatment regimens. A side-by-side analysis with similar studies could strengthen the argument for the chosen regimen.

Addressing Limitations: The discussion briefly mentions the study's urban focus and operational challenges. A more detailed exploration of how these factors might limit the study's applicability in rural or different socio-economic settings would be beneficial.

Limitations

Generalizability: The study’s focus on urban Dhaka limits its broader applicability. Discussing how these findings might translate to rural settings or countries with different healthcare infrastructures would be useful.

Diagnostic Methods: The study's reliance on clinical diagnosis without confirmatory tests like the Interferon-gamma Release Assay or Tuberculin Skin Test limits its validity. Discussing the potential implications of this limitation on the study's conclusions would be important.

Conclusion

Future Implications: The conclusion summarizes the findings but could be more explicit in suggesting practical applications for TB programs and strategies for scaling up the regimen, especially in resource-limited settings like Bangladesh.

More Specific Comments:

Main Research Question

  • The manuscript focuses on a specific and highly relevant question: evaluating the uptake and completion of a 12-dose, weekly isoniazid-rifapentine regimen (3HP) for TB prevention in a community setting. However, it could delve deeper into understanding the factors influencing these uptake and completion rates. For instance, it mentions that treatment completion was higher among females and those with more education and income, but it does not thoroughly explore why these demographics have higher completion rates. Such an exploration could provide more actionable insights for TB prevention programs.

Originality and Relevance

  • The study's focus on a shorter TB preventive treatment regimen is both original and highly relevant in the field of TB control. Given the global burden of TB and the challenge of long-duration treatments, the move to a shorter regimen like 3HP is significant. This study adds practical insights for high TB burden countries like Bangladesh.

Contribution to the Subject Area

  • The research adds to the evidence supporting shorter TB regimens, showing high completion rates (97%) in an urban Bangladeshi setting. This finding is critical as it suggests a feasible alternative to the traditional six-month regimen, which could improve adherence and reduce the burden on healthcare systems. However, the study could benefit from a more direct comparison with existing research on 3HP and other regimens to highlight its unique contributions or findings.

Methodological Improvements

  • The reliance on self-reported treatment adherence is a significant limitation. This method could introduce bias, as participants might overreport adherence. An objective measure, such as pill counting or medical record analysis, could strengthen the study's findings.
  • The lack of a control group using the standard 6-month IPT regimen makes it challenging to compare the effectiveness of the 12-dose regimen directly. Including such a control group could provide a more comprehensive understanding of the regimen's effectiveness.

Consistency of Conclusions

  • The conclusions about the high completion rate and low adverse event rate are consistent with the presented data. However, the conclusions might overstate the regimen's effectiveness without acknowledging the lack of a control group and the reliance on self-reported data.

Appropriateness of References:

  • The manuscript references studies relevant to TB preventive treatments. Still, it could benefit from including a broader range of studies, especially those comparing different regimens, to provide a more comprehensive context for its findings.

Comments on Tables and Figures:

  • The tables provide detailed socio-demographic data and information on adverse events. The table could benefit from better formating

Additional Comments:

  • The study's urban focus is a strength, given the urban concentration of TB cases. However, discussing the findings' applicability to rural settings, where healthcare access and patient behavior might differ significantly, would be beneficial.
  • The lack of confirmatory tests for latent TB infection (LTBI) is a limitation. The study could discuss how this affects the interpretation of the findings and their applicability in settings where LTBI tests are more commonly used.
  • Future research should explore the effectiveness of this regimen across different demographic groups, settings, and in comparison to the standard 6-month IPT regimen. This would help in understanding the broader applicability of the 3HP regimen.
Comments on the Quality of English Language

The English needs editing and improvement

Author Response

Reviewer #2: 

Abstract

Clarity and Conciseness: The abstract efficiently outlines the study's key aspects but includes some repetitive details, such as the specifics of the regimen and the study's setting. For instance, mentioning "12-dose, weekly isoniazid-rifapentine (3HP) TPT regimen" multiple times could be streamlined for brevity.

Thank you for your feedback on the abstract's clarity and conciseness. We acknowledge the redundancy in mentioning the "12-dose, weekly isoniazid-rifapentine (3HP) TPT regimen" multiple times. We have revised the abstract to eliminate repetitive details and streamline the information for brevity while ensuring that the key aspects of the study are still effectively communicated.

Introduction

Depth of Background Information: While the introduction highlights the global and national TB context, it could be improved by discussing specific challenges in Bangladesh, such as resource limitations or cultural barriers to TB prevention.

Acknowledging your suggestion, we have enriched the introduction to include specific challenges faced in Bangladesh regarding TB prevention. This includes resource limitations, cultural barriers, and unique epidemiological factors that affect TB control strategies (Lines -64-67).

Rationale for the Study: The introduction explains the choice of the 12-dose regimen but lacks comparative data on its efficacy against other regimens used in Bangladesh. A scenario where another regimen may have been less effective in a similar setting could illustrate the need for this study.

We appreciate the feedback on the rationale for the study from the reviewer. However, in the introduction (Lines 58-86), we referenced the current coverage of Isoniazid Preventive Treatment (IPT) in Bangladesh, which stands at 51%. This relatively low coverage emphasizes the need for alternative strategies/regimens. Our study focuses on the World Health Organization-endorsed shorter regimen, 3HP (12-dose, weekly isoniazid-rifapentine), which aligns with the United Nations high-level meeting (UNHLM) targets on TPT (among all age groups). The rationale is to assess whether the 3HP regimen can achieve better uptake and completion rates compared to the existing IPT approach, thereby contributing to more effective TB prevention in line with global health targets.

Methods

Study Design and Population: The method section is comprehensive but lacks detailed justification for excluding specific groups like pregnant women. Elaborating on the potential risks or ethical considerations for these exclusions would enhance understanding.

We appreciate your feedback on the method section. We agree that further elaboration on the exclusion of specific groups, such as pregnant women, is warranted. This exclusion is based on the potential risks associated with the use of rifapentine in pregnancy, which is not well studied and may pose unknown risks to fetal health. Additionally, ethical considerations dictate the need to avoid exposing pregnant women to any potential harm, especially when the safety profile of a treatment regimen in this demographic is not thoroughly established. We have updated the manuscript to include these justifications (Lines: 119 -123).

Data Collection Process: The manuscript details participant enrollment and treatment monitoring. However, it could describe the specific methods used to ensure adherence, such as digital tracking tools or patient diaries.

Thank you for the valuable suggestion. We appreciate the opportunity to provide further clarity on our data collection process. We have updated the text in the manuscript to better describe how we ensured adherence to the 3HP regimen (Lines: 164-165).

Results

Presentation of Findings: The results section clearly presents completion rates and demographic data. Providing confidence intervals for these rates would offer a more precise understanding of the data's reliability.

Thank you for the suggestion. We acknowledge the importance of providing confidence intervals to enhance the precision of our findings. We have incorporated confidence intervals for completion rates (Lines: 254-262) and in Table 3. However, we limited ourselves to basic statistical analysis for the demographic data.

Subgroup Analysis: The manuscript indicates different completion rates across demographics but doesn't delve into why these differences might exist. Analyzing factors like socioeconomic status or education level could provide deeper insights.

Thank you for the suggestion. We have performed bivariate and multivariable logistic regression examining the association between TPT completion with 3HP and demographic characteristics and clinical factors. The multivariable model reveals that the TPT completion was higher in contacts aged 15 years or more (OR 1.5; 95% CI 1.1-2.0; p 0.043); female contacts (OR 1.7; 95% CI 1.3-2.1; p 0.009); contacts with higher education (OR 1.4; 95% CI 1.1-1.9; p 0.044), contacts with high HH income (OR 1.5; 95% CI 1.0-2.1; p 0.047), contacts those with no comorbidities (OR 1.7; 95% CI 1.1-2.2; p 0.046), and who have not had experienced any adverse events while on 3HP  regimen (OR 1.6; 95% CI 1.2-2.1; p 0.009). This is included in the text in the result section (Lines: 254-262) and in Table 3. 

Discussion

Comparative Analysis: While the discussion puts the study in context, it lacks a detailed comparison with other studies using different TB treatment regimens. A side-by-side analysis with similar studies could strengthen the argument for the chosen regimen.

Thank you for your valuable feedback. We appreciate your suggestion to include a more detailed comparative analysis with studies using different TB treatment regimens. We agree that such a comparison can strengthen the argument for the chosen regimen and provide a broader perspective on its effectiveness. However, the main objective of this study was to evaluate the 3HP regimen, which was the newer, shorter TPT regimen endorsed by the WHO at the time. We aimed to assess its acceptance, including uptake and completion rates, in comparison to the 6-month IPT regimen that was routinely used under the programmatic setting of Bangladesh. We hope this clarifies the purpose of our study and the rationale behind our approach. Thank you again for your feedback.

Addressing Limitations: The discussion briefly mentions the study's urban focus and operational challenges. A more detailed exploration of how these factors might limit the study's applicability in rural or different socio-economic settings would be beneficial.

Thank you for your suggestion. We have substantiated the limitation section in our manuscript to provide a more comprehensive context for our findings. We believe that this will indeed help balance the findings and their interpretation by clearly outlining the study's constraints and the need for further research. Your feedback has been valuable in enhancing the quality of our manuscript (Lines: 352-378).

Limitations

Generalizability: The study’s focus on urban Dhaka limits its broader applicability. Discussing how these findings might translate to rural settings or countries with different healthcare infrastructures would be useful.

Thank you for your comment. We acknowledge the importance of discussing the applicability of our findings to rural settings. While our study focused on urban settings in Dhaka, we recognize the need to consider the differences in healthcare access and patient behavior in rural areas. We agree that future research should explore the feasibility and implementation of TPT in rural settings to provide a more comprehensive understanding of TPT's applicability across different contexts. To address this concern, we have added a brief discussion in the limitations section of our manuscript, highlighting the urban focus of our study and its potential limitations in representing rural populations (Lines: 354-363).

Diagnostic Methods: The study's reliance on clinical diagnosis without confirmatory tests like the Interferon-gamma Release Assay or Tuberculin Skin Test limits its validity. Discussing the potential implications of this limitation on the study's conclusions would be important.

Thank you for raising the issue of the lack of confirmatory tests for LTBI in our study. We have already acknowledged this limitation. In our study, we did not employ confirmatory tests for LTBI due to practical constraints and the high TB burden setting. While our study did not employ confirmatory tests for LTBI, it's important to acknowledge that in settings where LTBI tests are more commonly used, the interpretation of our findings may need to be considered with caution. The absence of LTBI confirmation in our study could potentially result in some misclassification of LTBI status, and this may impact the generalizability of our results to settings with routine LTBI testing. In such settings, the decision to initiate TPT might be guided by a combination of clinical evaluation, LTBI test results, and individual risk factors. Therefore, the applicability of our findings to areas with robust LTBI testing should be made with an awareness of this limitation. We have included this in the text to substantiate the limitation section (Lines: 371-380).

Conclusion

Future Implications: The conclusion summarizes the findings but could be more explicit in suggesting practical applications for TB programs and strategies for scaling up the regimen, especially in resource-limited settings like Bangladesh.

Thank you for your feedback, and we appreciate your suggestion regarding the future implications of our study. We agree that the conclusion can be more explicit in suggesting practical applications for TB programs and strategies for scaling up the regimen, particularly in resource-limited settings like Bangladesh. In response to your comment, we have further substantiated the conclusion section to provide clearer and more explicit recommendations for TB programs and strategies for scaling up the 3HP regimen (Lines: 386-393).

 More Specific Comments:

Main Research Question

  • The manuscript focuses on a specific and highly relevant question: evaluating the uptake and completion of a 12-dose, weekly isoniazid-rifapentine regimen (3HP) for TB prevention in a community setting. However, it could delve deeper into understanding the factors influencing these uptake and completion rates. For instance, it mentions that treatment completion was higher among females and those with more education and income, but it does not thoroughly explore why these demographics have higher completion rates. Such an exploration could provide more actionable insights for TB prevention programs.

Thank you for your thoughtful comment. We agree that understanding the factors influencing uptake and completion rates is crucial for TB prevention programs. While our study did observe that treatment completion was higher among females and those with more education and income, a more in-depth exploration of the underlying reasons behind these demographic differences could indeed provide actionable insights. In our ongoing efforts, we plan to conduct further research to delve deeper into these factors. This will involve qualitative research methods, such as interviews and focus group discussions, to gain a better understanding of the motivations and barriers faced by different demographic groups. We aim to provide more comprehensive insights that can inform targeted interventions to improve TB prevention program outcomes.

Originality and Relevance

  • The study's focus on a shorter TB preventive treatment regimen is both original and highly relevant in the field of TB control. Given the global burden of TB and the challenge of long-duration treatments, the move to a shorter regimen like 3HP is significant. This study adds practical insights for high TB burden countries like Bangladesh.

Thank you for recognizing the significance of our study's focus on a shorter TB preventive treatment regimen. We appreciate your positive feedback on the relevance of our research in the field of TB control, especially in high TB burden countries like Bangladesh. Your encouraging comments motivate us to continue our efforts to improve TB prevention and control strategies.

Contribution to the Subject Area

  • The research adds to the evidence supporting shorter TB regimens, showing high completion rates (97%) in an urban Bangladeshi setting. This finding is critical as it suggests a feasible alternative to the traditional six-month regimen, which could improve adherence and reduce the burden on healthcare systems. However, the study could benefit from a more direct comparison with existing research on 3HP and other regimens to highlight its unique contributions or findings.

Thank you for your valuable comment. We agree that a more direct comparison with existing research on 3HP and other regimens would enhance the manuscript. We will include a discussion section that highlights the unique contributions and findings of our study in relation to existing research on 3HP and other regimens. Your input is greatly appreciated in strengthening our manuscript.

Methodological Improvements

  • The reliance on self-reported treatment adherence is a significant limitation. This method could introduce bias, as participants might overreport adherence. An objective measure, such as pill counting or medical record analysis, could strengthen the study's findings.

Thank you for the valuable suggestion. We appreciate the opportunity to provide further clarity on measuring self-reported treatment adherence. We have updated the text in the manuscript to better describe how we measured adherence to the 3HP regimen (Lines: 163-166).

  • The lack of a control group using the standard 6-month IPT regimen makes it challenging to compare the effectiveness of the 12-dose regimen directly. Including such a control group could provide a more comprehensive understanding of the regimen's effectiveness.

We acknowledge the issue raised by the reviewer to not select a true comparator group for the effectiveness of the 3HP regimen. Through this study, 3HP was first introduced in Bangladesh, and we did not have a choice to compare this with a comparator group on TPT other than the 6-month IPT. However, we have substantiated the limitation section to highlight the study limitation around not having a true comparator (Lines: 364-371).

Consistency of Conclusions

  • The conclusions about the high completion rate and low adverse event rate are consistent with the presented data. However, the conclusions might overstate the regimen's effectiveness without acknowledging the lack of a control group and the reliance on self-reported data.

Thank you for your suggestion. We have substantiated the limitation section in our manuscript to provide a more comprehensive context for our findings. We believe that this will indeed help balance the conclusion and its interpretation by clearly outlining the study's constraints and the need for further research. Your feedback has been valuable in enhancing the quality of our manuscript.

Appropriateness of References:

  • The manuscript references studies relevant to TB preventive treatments. Still, it could benefit from including a broader range of studies, especially those comparing different regimens, to provide a more comprehensive context for its findings.

We acknowledge the point raised by the reviewer on references. However, the main objective of this study was to evaluate the 3HP regimen, which was the newer, shorter TPT regimen endorsed by the WHO at the time. We aimed to assess its acceptance, including uptake and completion rates, in comparison to the 6-month IPT regimen that was routinely used under the programmatic setting of Bangladesh. Hence, we did not include the references that reflect the comparison of different regimens.

Comments on Tables and Figures:

  • The tables provide detailed socio-demographic data and information on adverse events. The table could benefit from better formatting

As suggested, we have formatted the tables.

Additional Comments:

  • The study's urban focus is a strength, given the urban concentration of TB cases. However, discussing the findings' applicability to rural settings, where healthcare access and patient behavior might differ significantly, would be beneficial.

Thank you for your comment. We acknowledge the importance of discussing the applicability of our findings to rural settings. While our study focused on urban settings in Dhaka, we recognize the need to consider the differences in healthcare access and patient behavior in rural areas. We agree that future research should explore the feasibility and implementation of TPT in rural settings to provide a more comprehensive understanding of TPT's applicability across different contexts. To address this concern, we have added a brief discussion in the limitations section of our manuscript, highlighting the urban focus of our study and its potential limitations in representing rural populations (Lines: 354-364).

  • The lack of confirmatory tests for latent TB infection (LTBI) is a limitation. The study could discuss how this affects the interpretation of the findings and their applicability in settings where LTBI tests are more commonly used.

Thank you for raising the issue of the lack of confirmatory tests for LTBI in our study. We have already acknowledged this limitation. In our study, we did not employ confirmatory tests for LTBI due to practical constraints and the high TB burden setting. While our study did not employ confirmatory tests for LTBI, it's important to acknowledge that in settings where LTBI tests are more commonly used, the interpretation of our findings may need to be considered with caution. The absence of LTBI confirmation in our study could potentially result in some misclassification of LTBI status, and this may impact the generalizability of our results to settings with routine LTBI testing. In such settings, the decision to initiate TPT might be guided by a combination of clinical evaluation, LTBI test results, and individual risk factors. Therefore, the applicability of our findings to areas with robust LTBI testing should be made with an awareness of this limitation. We have included this in the text to substantiate the limitation section (Lines: 371-380).

  • Future research should explore the effectiveness of this regimen across different demographic groups, settings, and in comparison to the standard 6-month IPT regimen. This would help in understanding the broader applicability of the 3HP regimen.

We appreciate your suggestion. Indeed, understanding the effectiveness of the 3HP regimen across various demographic groups and settings is crucial for assessing its broader applicability. Additionally, comparing it with the standard 6-month IPT regimen is an important avenue for research. In our discussion, have emphasized further the need for further research to delve into these aspects, as they can provide valuable insights into the optimal use of the 3HP regimen and its potential advantages in specific populations or settings (Lines: 364-371).

Reviewer 3 Report

Comments and Suggestions for Authors

The article “Uptake and completion of tuberculosis preventive treatment 2 using 12-dose, weekly isoniazid-rifapentine regimen in Bangla- 3 desh: a community-based implementation study”  is a very interesting paper that contemplates the possibility of an adjustment to the regimen. treatment with isoniazid-rifapentine.

The number of patients included is adequate, as is the methodological strategy.

Currently, the search for new treatment regimens and the use of anti-inflammatory drugs continues to be a great challenge worldwide and these studies are of great progress, although preliminary.

It is recommended to improve the quality of figure 1.

Author Response

Reviewer #3: 

The article “Uptake and completion of tuberculosis preventive treatment 2 using 12-dose, weekly isoniazid-rifapentine regimen in Bangla- 3 desh: a community-based implementation study”  is a very interesting paper that contemplates the possibility of an adjustment to the regimen. treatment with isoniazid-rifapentine.

The number of patients included is adequate, as is the methodological strategy.

Currently, the search for new treatment regimens and the use of anti-inflammatory drugs continues to be a great challenge worldwide and these studies are of great progress, although preliminary.

Thank you for your positive feedback on the article. We appreciate your recognition of the study's importance in exploring potential adjustments to the treatment regimen, the study's methodology and number of patient inclusion, and its contribution to the ongoing global efforts to find effective treatment regimens, although it represents preliminary progress in this area.

It is recommended to improve the quality of figure 1.

As suggested, we have tried to improve the quality of Figure 1.

Reviewer 4 Report

Comments and Suggestions for Authors

This is an important manuscript guiding NTP on the scaling up TPT with 3HP with a high completion rate, and can also guide countries with similar contexts. Minor comments:

Line 55: how much was the increase in notification?

Line 61: data on coverage of HIV and completion of TPT in children and HIV can be added. 

Line 75: add the meaning of HH as this is the first time is mentioned.

Line 102: how the facilities included in the study were selected?

Line 114: What is the profile of FS?

Line 197: Does this mean that 1804 patients who had an indication of CXR were tested? Not clear if all with indication of CXR accessed this test, this is an issue in most countries as CXR is not always available and patients have to pay in some cases.

Line 245: should be "high" and not low-income

Line 302: how side effects were managed? free of charge?

Line 330: CXR is more sensitive and coverage here was not 100%, this should be a lesson learned for advocating using CXR as a first test as we know that are subclinical or asymptomatic patients.

Author Response

Reviewer #4: 

This is an important manuscript guiding NTP on the scaling up TPT with 3HP with a high completion rate, and can also guide countries with similar contexts.

Thank you for recognizing the importance of our manuscript in guiding TB prevention efforts. We appreciate your feedback and have addressed the minor comments to ensure the manuscript's clarity and completeness.

Minor comments:

Line 55: how much was the increase in notification?

The notification rate has been increasing by approximately 5% annually since 2001, starting from 59 per 100,000 people for all forms of TB and reaching 161 per 100,000 people for all forms of TB in 2018. We have included this in the text for further clarification (Line: 57).

Line 61: data on coverage of HIV and completion of TPT in children and HIV can be added. 

Thanks for the suggestions on adding coverage and completion among patients living with HIV and children. We have included the coverage and completion rate of IPT among children in the text (Lines: 62-64). However, we could not find the TPT data on HIV patients in the report.

Line 75: add the meaning of HH as this is the first time is mentioned.

This has been abbreviated in its first appearance as suggested by the reviewer (Line: 81).

Line 102: how the facilities included in the study were selected?

Thanks to the reviewer for the clarification request. We selected all the 12-NTP linked facilities in the three (Zones 3,4 and 5) of Dhaka South City Corporation. We have incorporated this in the text for further clarification (Lines: 100 and 109).

Line 114: What is the profile of FS?

Thank you for seeking clarification. The field supervisors were trained health workers with prior experience in working on TB and recruited by the study, responsible for enumeration of HH contacts, and bringing HH contacts to health facilities for screening. We have now included this information in the manuscript for clarity (Lines: 126-127).

Line 197: Does this mean that 1804 patients who had an indication of CXR were tested? Not clear if all with indication of CXR accessed this test, this is an issue in most countries as CXR is not always available and patients have to pay in some cases.

Thanks to the reviewer for highlighting this. The study offered verbal symptomatic screening and CXR to all contacts (n=2,149) who visited the health facility upon invitation. However, 1,804 (84%) of the contacts completed an evaluation with CXRs. The remaining, 345 (16%) did not attend the CXR screening site though it was offered free from the project. We have rephrased the method section to further clarify this (Lines: 128-138).

Line 245: should be "high" and not low-income

Thank you for pointing out the error in line 245. It should indeed be "high HH income," and we appreciate your attention to detail. This correction has been made in the revised manuscript to accurately describe the economic status of the study participants (Line: 260).

Line 302: how side effects were managed? free of charge?

Thanks to the reviewer for the clarification requested on the management of the side effects. The side effects were managed by the physicians free of cost. We have described this in the methods section (Lines: 166-179).

Line 330: CXR is more sensitive and coverage here was not 100%, this should be a lesson learned for advocating using CXR as a first test as we know that are subclinical or asymptomatic patients.

We acknowledge the value of chest X-ray due to its higher sensitivity for TB detection, especially to identify subclinical or asymptomatic patients. Our study did not achieve 100% coverage, which highlights the need to advocate CXR usage in early TB detection strategies. As suggested, we have included this in the discussion section (Lines: 349-352).

Round 2

Reviewer 2 Report

Comments and Suggestions for Authors

The authors have addressed most of my comments from the previous review. The document is good to go. However, given that some of the patients were also initiated on IPT within the study based on the study flowchart, I will like to see the addition of NTP recommendations for IPT and TPT in the country. This will be important for readers to understand while both medications were given in the study